

# An ecological transcriptome approach to capture the molecular and physiological mechanisms of mass flowering in *Shorea curtisii*

Ahmad Husaini Suhaimi[1], Masaki J. Kobayashi[2], Akiko Satake[3], Ching Ching Ng[1], Soon Leong Lee[4], Norwati Muhammad[4], Shinya Numata[5], Tatsuya Otani[6], Toshiaki Kondo[7], Naoki Tani[2,8] and Suat Hui Yeoh[1]

[1] Institute of Biological Sciences, Faculty of Science, Universiti Malaya, Kuala Lumpur, Malaysia
[2] Forestry Division, Japan International Research Center for Agricultural Sciences, Tsukuba, Ibaraki, Japan
[3] Department of Biology, Faculty of Science, Kyushu University, Fukuoka, Japan
[4] Forestry Biotechnology Division, Forest Research Institute Malaysia, Selangor, Malaysia
[5] Department of Tourism Science, Tokyo Metropolitan University, Tokyo, Japan
[6] Shikoku Research Center, Forestry Research and Management Organization, Kochi, Japan
[7] Bio-Resources and Utilization Division, Japan International Research Center for Agricultural Sciences, Tsukuba, Ibaraki, Japan
[8] Faculty of Life and Environmental Sciences, University of Tsukuba, Tsukuba, Ibaraki, Japan

Corresponding authors
Naoki Tani, ntani@affrc.go.jp
Suat Hui Yeoh, suathui.yeoh@gmail.com

## ABSTRACT

Climatic factors have commonly been attributed as the trigger of general flowering, a unique community-level mass flowering phenomenon involving most dipterocarp species that forms the foundation of Southeast Asian tropical rainforests. This intriguing flowering event is often succeeded by mast fruiting, which provides a temporary yet substantial burst of food resources for animals, particularly frugivores. However, the physiological mechanism that triggers general flowering, particularly in dipterocarp species, is not well understood largely due to its irregular and unpredictable occurrences in the tall and dense forests. To shed light on this mechanism, we employed ecological transcriptomic analyses on an RNA-seq dataset of a general flowering species, *Shorea curtisii* (Dipterocarpaceae), sequenced from leaves and buds collected at multiple vegetative and flowering phenological stages. We assembled 64,219 unigenes from the transcriptome of which 1,730 and 3,559 were differentially expressed in the leaf and the bud, respectively. Differentially expressed unigene clusters were found to be enriched with homologs of *Arabidopsis thaliana* genes associated with response to biotic and abiotic stresses, nutrient level, and hormonal treatments. When combined with rainfall data, our transcriptome data reveals that the trees were responding to a brief period of drought prior to the elevated expression of key floral promoters and followed by differential expression of unigenes that indicates physiological changes associated with the transition from vegetative to reproductive stages. Our study is timely for a representative general flowering dipterocarp species that occurs in forests that are under the constant threat of deforestation and climate change as it pinpoints important climate sensitive and flowering-related homologs and offers a glimpse into the cascade of gene expression before and after the onset of floral initiation.

## INTRODUCTION

General flowering (GF) is a unique intermittent flowering phenomenon that occurs at irregular intervals of 1–10 years in Southeast Asian mixed-lowland dipterocarp forests (*Ashton, Givnish & Appanah, 1988*; *Chen et al., 2018*). During GF, dozens of plant families, including Dipterocarpaceae, exhibit synchronous flowering (*Appanah, 1993*; *Ashton, Givnish & Appanah, 1988*) which lasts for about four weeks. Typically, GF is followed by mast fruiting (*Appanah, 1985*; *Sakai et al., 1999*), resulting in a massive pulse of resources in the form of seeds and fruits that are consumed by nectarivorous and frugivorous fauna (*Ashton, Givnish & Appanah, 1988*; *Janzen, 1974*). The interval and magnitude of GF are proposed to be influenced by climate extremes such as El Nino-Southern Oscillation (ENSO), which leads to fluctuations in temperatures, drops in cumulative rainfalls and drought events (*Sakai et al., 2006*). The unpredictable nature of GF presents challenges for seed collection for the conservation of GF dipterocarp species (*Kettle, 2010*), which are among the most abundant trees in the tropical rainforests (*Ashton, 1988*). Hence, determining the cues that govern flowering timing of GF dipterocarps is critical for conservation management of these species, in a region that is constantly threatened by deforestation.

The absence of clear seasonality in the aseasonal tropics poses difficulties when it comes to identifying the proximate cues and underlying physiological and molecular mechanisms of GF dipterocarps. While photoperiod or day length is a key factor in the reproductive initiation of temperate plants (*Andres & Coupland, 2012*), the minimal day length variation in the tropics (*Janzen, 1967*) suggests that this may not be the case for tropical plants. Studies in *Citrus* spp. (*Agusti et al., 2022*; *Khan et al., 2022*; *Li et al., 2017*) have shed some light on the trigger of flowering for plants in this region. Unlike their subtropical counterparts, which rely on low temperatures as floral cue, tropical citrus has been found to initiate flowering after a brief period of drought (*Agusti et al., 2022*; *Khan et al., 2022*). These observations align with earlier reports that drought can mediate or accelerate flowering in various plants including *Arabidopsis thaliana* (*Takeno, 2016*) and wheat (*Isidro et al., 2011*). The aforementioned studies in *Citrus* spp. have also unveiled potential molecular mechanisms linking drought signals to flowering (*Khan et al., 2022*; *Li et al., 2017*). These studies found elevated expression levels of key proteins in floral regulation, including GIGANTEA (GI), following an increase in endogenous abscisic acid (ABA) levels in these trees (*Khan et al., 2022*; *Li et al., 2017*). Subsequently, there was an upregulation of the major floral promoter, FLOWERING LOCUS T (FT), expression in leaves and buds, indicating floral initiation. *Li et al. (2017)* also reported a decrease in gibberellin (GA) levels in lemon buds prior to flowering, aligning with earlier studies that reported GA as a floral repressor in woody plants (reviewed in *Wilkie, Sedgley & Olesen (2008)*). While these trees are not GF species, the knowledge on the physiological and molecular aspects in floral regulation of these trees could provide an insight into flowering

time control in dipterocarps. The question of whether a similar mechanism is employed in GF dipterocarps remains to be investigated.

Progress to understand the floral regulation of dipterocarps has been hindered by limited sample availability for frequent sample collection and challenges associated with accessing the dense and tall canopies of tropical forests. Despite sampling challenges, several studies have utilized gene expression and ecological data to infer the environmental factors regulating floral initiation in GF dipterocarps. For example, transcriptome sequencing of bud samples from *Shorea beccariana*, coupled with rainfall data, supported drought as a significant floral trigger (*Kobayashi et al., 2013*). In a separate study, the expression of two key flowering genes, *FLOWERING LOCUS T* (*FT*) and *LEAFY* (*LFY*), in two *Shorea* species, *Shorea curtisii* and *Shorea leprosula,* measured using real-time PCR, showed a strong correlation with floral initiation (*Yeoh et al., 2017*). The same study also showed that the synergistic effect of drought and low temperature best predicted the occurrence of flowering in the study period (*Yeoh et al., 2017*). This report was followed by our recent research using RNA-seq data from *S. curtisii* leaf samples collected at three time points, including before and after floral initiation (*Suhaimi et al., 2023*). In the study, homologs of genes associated with drought response were found to have a significantly higher representation among genes that were differentially expressed during flowering time (*Suhaimi et al., 2023*), thus supported the involvement of drought in floral initiation of GF dipterocarps. The studies thus far, nevertheless important, were conducted on a small number of key flowering genes (*Yeoh et al., 2017*) or a single tissue (*Kobayashi et al., 2013*; *Suhaimi et al., 2023*). Since physiological changes during flowering span across multiple organs (*Parcy, 2005*) with both leaves and buds playing direct roles in the production and transportation of proteins that regulate floral initiation, a concurrent study of both the organs is needed to capture the missing link and gain a comprehensive view of the gene expression dynamics that occur between the two organs. Moreover, leveraging information available on model species, *A. thaliana* and tropical citrus, coupled with the ecological transcriptome approach that combines transcriptome and meteorological data (*Richards et al., 2009*), offer a means to elucidate the cues and regulatory mechanism of flowering in GF dipterocarp species, thus enabling the prediction of flowering time.

In our current study, we aimed to capture the transcriptomic changes in a dipterocarp species, *S. curtisii*, across a flowering year and infer the physiological and molecular mechanisms leading to flowering in this species. To achieve this objective, we utilized the transcriptome of both leaf and bud tissues, and more than double the number of time points compared to our previous study (*Suhaimi et al., 2023*). Our current study provided a greater resolution of the molecular events leading to flowering in a GF dipterocarp species. We identified and characterized differentially expressed unigenes (DEUs) by comparing them to publicly available databases and *A. thaliana* gene sets that are associated with responses to biotic and abiotic stresses, nutrient levels, and hormonal treatments. We also integrated meteorological records to uncover the molecular pathways interacting with the environmental signals and finally discussed the possible mechanisms involved in the floral regulation of this GF dipterocarp species.

## MATERIALS & METHODS

In the current study, we leveraged transcriptome sequencing analysis to elucidate the expression dynamics of unigenes in *S. curtisii* over a GF season (Fig. S1). After *de novo* assembly, we annotated the constructed non-redundant unigenes using selected protein databases. Then, transcript quantification and principal component analysis (PCA) were performed. Subsequently, differential expression analysis was conducted and the DEUs were clustered based on their expression patterns. Following this, enrichment analyses of genes that have been reported to participate in responses to various nutrients and stresses in the model plants were performed, initially on the DEUs followed by the DEU clusters. We further characterized the clusters of DEUs using GO and KEGG databases. Finally, these results were compared with available meteorological data to obtain some insights into the interplay between gene expression patterns and environmental cues.

### Study species and sample collection

The species selected for this study, *S. curtisii* (Dipterocarpaceae), and other trees from the genus *Shorea* section *Mutica*, have been documented as reliable indicators of GF (*Ashton, Givnish & Appanah, 1988*). *Shorea* spp. can be found occupying ridges of dipterocarp forests in the Malay Peninsula ranging from 300–800 m above sea level (*Appanah & Chan, 1981*).

Due to the unique challenges associated with sampling from the canopy of these trees (∼40 m above the ground canopy), which requires the setup of ladders along the main trunk, we were limited to accessing and collecting samples from canopies of only two *S. curtisii* individuals. These trees, designated as C1 and C2, had diameters at breast height of 72 cm and 89 cm, respectively (*Yeoh et al., 2017*). The trees are located at Semangkok Forest Reserve (2°58′N, 102°18′E, 340–450 m above sea level) in the state of Selangor Darul Ehsan, Peninsular Malaysia (Fig. S2 A). The permission to conduct this research in the forest reserve was granted by the Selangor State Forest Department. The forest reserve consists of a 6-ha (200 m × 300 m) hill dipterocarp forest conservation plot, and another 5.4-ha (140 m × 400 m) plot established on a selectively logged area (*Tani et al., 2015*; *Yagihashi et al., 2010*). Daily temperature and precipitation data were collected from Hospital Kuala Kubu Bharu meteorological station and Kampung Pertak hydrological station, respectively (Fig. S2B), and were previously reported in *Yeoh et al. (2017)*. The cumulative rainfall data (defined as 30-d moving rainfall total) was calculated from the daily rainfall record. Drought is defined as <40 mm cumulative rainfall (*Sakai et al., 2006*).

To obtain gene expression profiles of the study species over a flowering period, we collected leaves and buds sampled from the top canopies of the two *S. curtisii* trees around noon (Table S1). The concurrent use of both tissues is essential to understand the gene expression dynamics between them, as both tissues are involved in the production and transportation of floral regulatory proteins. These samples, which are part of the collection described in *Yeoh et al. (2017)*, consisted of eight time points (TPs): 08 July 2013 (TP-A), 17 October 2013 (TP-B), 18 December 2013 (TP-C), 14 February 2014 (TP-D), 21 March 2014 (TP-E), 02 April 2014 (TP-F), 14 May 2014 (TP-G), and 17 June 2014 (TP-H). The TPs were selected to represent different flowering stages of the trees, which are categorized based on

morphological characters of the buds observed during sampling. In this study, the trees were classified into four stages: (i) vegetative stage, characterized by vegetative buds that would differentiate into leaves, (ii) inflorescence stage, indicated by the presence of inflorescence, (iii) reproductive/flowering stage, marked by the presence of some reproductive buds that would differentiate into flowers, and (iv) abortion/fruiting stage, in which the flowers either developed into fruits or fell off (Fig. S3 and Table S1). Multiple time points were selected for the vegetative stage to capture changes in gene expression leading up to the flowering period. After collection, the samples were immediately submerged in RNAlater solution (Ambion, Austin, TX, USA) and stored at −80 °C until RNA extraction. Using RNASeqPower package (*Hart et al., 2013*) in R v4.0.3 (*R Core Team, 2018*), the statistical power of the experimental design was calculated to be 0.38 for a sample size of two (the number of biological replicates used in the study).

## Transcriptome sequencing and annotation

Total RNA was extracted from leaf and bud samples using CTAB method described by *Kobayashi et al. (2013)*. The extracted RNA was used to create paired-end cDNA libraries as per manufacturer's protocol (Illumina, Foster City, CA, USA) followed by RNA-seq using Illumina HiSeq 4000 Sequencer (Illumina). Low quality bases and traces of Illumina adapters were removed using Trimmomatic v0.36 (*Bolger, Lohse & Usadel, 2014*) prior to *de novo* assembly using Trinity v2.8.5 (*Grabherr et al., 2011*). To minimize transcript redundancy, only transcripts with complete open reading frames (ORFs) as predicted by TransDecoder v5.5.0 (http://transdecoder.github.io) were retained. The number of transcripts was further reduced by clustering highly similar protein sequences using CD-HIT v4.8.1 (*Fu et al., 2012*) (parameters: -n 5 -c 1). The completeness of the assembly was assessed using BUSCO v5.2.1 (*Manni et al., 2021*). Unless otherwise stated, all bioinformatic analyses in this study were conducted using default parameters.

The non-redundant transcripts, henceforth referred to as unigenes, were annotated by querying against the proteome of *A. thaliana* (*Cheng et al., 2017*) using BLASTx v2.10 (*Camacho et al., 2009*) with *E*-value cut-off: 1E–10. The annotated unigenes were compared to the *A. thaliana* flowering genes database (*Bouché et al., 2016*) to identify their homologs in *S. curtisii*. Translated unigenes were searched against UniProt database using BLASTp v2.10 (*Camacho et al., 2009*) with *E*-value cut-off: 1E–05, as well as Pfam v32.0 (*El-Gebali et al., 2018*) and PANTHER v14.1 (*Thomas et al., 2003*) databases using InterProScan v5.36 (*Jones et al., 2014*). The unigenes were assigned Gene Ontology (GO) terms (*Ashburner et al., 2000*) associated with the earlier BLASTx matches and visualized using WEGO online tool (*Ye et al., 2018*). The unigenes were also queried against KEGG database (*Kanehisa & Goto, 2000*) using BlastKOALA online tool (*Kanehisa, Sato & Morishima, 2016*).

## Differential expression analysis and enrichment tests

Transcript quantification was conducted using Salmon v1.5.1 (*Patro et al., 2016*). To summarize the expression data, we performed a principal component analysis (PCA) using *pcaExplorer* package v2.22.0 (*Marini & Binder, 2019*) in R, which normalized the expression values using DESeq method (*Anders & Huber, 2010*) prior to PCA. Subsequent

bioinformatic analyses were performed on leaf and bud samples separately. We conducted differential expression analysis using DESeq2 software v1.32.0 (*Love, Huber & Anders, 2014*) by comparing all possible pairwise TPs (Table S2) to identify DEUs (defined as transcripts with absolute $\log_2$ fold change $\geq 1$ and false discovery rate (FDR) <1E–03). To identify significantly enriched GO terms ($P < 0.05$) in the DEUs, enrichment analysis was performed using topGO package v2.40.0 (*Alexa & Rahnenfuhrer, 2020*) in R. Enriched KEGG pathways (FDR <0.05) were identified using an online tool, KO-Based Annotation System (KOBAS) v3.0 (*Bu et al., 2021*). To validate the expression profiles of the unigenes, the expression of key flowering gene, *FT*, homologs was compared to relative expression values measured by qRT-PCR in a previous study in *S. curtisii* (*Yeoh et al., 2017*). We utilized *ggpubr* package in R to perform Spearman's correlation test and to plot linear regression scatter plots.

## Clustering and characterization of differentially expressed unigenes

To summarize the expression profiles of the DEUs in both leaf and bud tissues, the unigenes were clustered based on their expression patterns using *coseq* package (*Rau & Maugis-Rabusseau, 2018*) in R. The expression values were normalized using Trimmed Mean of M-values (TMM) method (*Robinson & Oshlack, 2010*), transformed using centered log ratio (*Aitchison, 1982*), and clustered using K-mean algorithm (*MacQueen, 1967*).

For the characterization of the DEUs, we followed the method used in the transcriptome study of *S. beccariana* (*Kobayashi et al., 2013*), which compared DEUs to *A. thaliana* transcriptome data and assumed that the regulation of homologs in both species is similar. This approach enabled functional annotations of DEUs with categories that are not covered by the GO database, such as "response to prolonged moderate drought" (*Kobayashi et al., 2013*). We compared the DEUs in *S. curtisii* to 35 gene sets of up- and downregulated genes in *A. thaliana* that were exposed to various external and endogenous factors (*Cao et al., 2006*; *Gould et al., 2006*; *Harb et al., 2010*; *Ma & Bohnert, 2007*; *Misson et al., 2005*; *Nemhauser, Hong & Chory, 2006*; *Osuna et al., 2007*; *Scheible et al., 2004*; see Table S3 for details). Fisher's exact tests with Bonferroni multiple testing correction were performed to determine the significance in the number of overlapping genes between DEUs and the *A. thaliana* gene sets (Fig. S4). When the number of DEUs that overlaps with the genes in the gene set is significant, we assumed that the *S. curtisii* trees used in this study experienced similar conditions as the *A. thaliana* from which the gene set was obtained.

To determine if the significantly tested *A. thaliana* gene sets showed specific expression patterns, we further compared the gene sets to DEU clusters using gene enrichment tests (Fig. S5 ; Tables S16–S23, S24–S30). The enrichment tests were performed using Fisher's exact tests with Bonferroni corrections. We assumed enriched gene sets had the same expression pattern as their significantly associated DEU clusters. Finally, the DEU clusters with significantly enriched gene sets were compared to the temperature and rainfall records to determine potential association between the transcriptome and meteorological data.

## RESULTS

### RNA sequencing and transcriptome assembly

The transcriptome profile of *S. curtisii* at different stages of flowering (vegetative, inflorescence, flowering, and fruiting/abortion) was captured by sequencing total RNA from leaves and buds at eight time points (TPs). The sequencing generated 1,595,323,292 reads with an average length of 150 bp which were assembled into 664,053 transcripts. After removing redundant transcripts, the resulting transcriptome assembly comprises 64,219 unigenes with an average length of 2,148 bp and N50 of 2,624 bp (Fig. S6). The GC content of the assembled transcriptome was 43.02%. The BUSCO analysis indicated an almost complete transcriptome assembly in which 95.5% of highly conserved proteins in the Embryophyta lineage were recovered (Table S4). Of the total unigenes, 58,722 (91.44%) were annotated with at least one of the public protein databases (Tabes S5–S7; Fig. S7).

We conducted PCA on normalized unigene expression data to obtain an overview of the variance in gene expression among samples. The analysis clustered the samples based on tissue type, individual tree, and TPs (Fig. 1A). Most of the variance (PC1: 34.71%) can be attributed to transcriptional differences between buds and leaves (Fig. 1A). Variation in transcriptional profiles between the two biological replicates was also high (PC2: 18.63%; Fig. 1A). In bud, 11.29% of the variation (PC3) distinguished majority of the bud samples at vegetative stages (TP-A–D) from the remaining bud samples (TP-E–H; Fig. 1B). The clustering of TP-E with TPs at the inflorescence (TP-F) and later stages (TP-G–H) suggests that the switch to inflorescence may have already begun although morphologically the buds at TP-E are still in vegetative stage. Interestingly, a number of the top contributing unigenes for PC3 (Table S10) are homologs of stress-responsive genes such as *EXPANSIN A15* (*Wieczorek et al., 2006*) and *WRKY33* (*Zheng et al., 2006*).

### Differential gene expression and clusters of differentially expressed unigenes

To identify candidate unigenes that are involved in floral regulation of *S. curtisii*, we conducted a differential expression analysis for pairwise comparison of all TPs. A total of 1,730 and 3,559 DEUs were identified in leaf and bud samples, respectively, with 789 unigenes shared between the two tissues (Tables S11 and S12). The differences in the number of DEUs between the tissues may be due to the regulation of organ-specific unigenes that function in floral morphogenesis, as previously reported in *A. thaliana* (*Smaczniak et al., 2017*). To summarize the expression profiles of the DEUs, K-mean clustering was performed, resulting in the identification of six DEU clusters with distinctive expression profiles in leaf transcriptomes (Fig. 2) and five DEU clusters in bud transcriptomes (Fig. 3).

### Differential expression of putative flowering-time unigenes

To gain insight into the floral regulatory pathways of *S. curtisii*, we first identified homologs of *A. thaliana* flowering-time genes from the annotated unigenes. Of the 51,654 unigenes that could be mapped to *A. thaliana* genes, 918 unigenes were homologous to 228 non-redundant flowering-time genes (Table 1 and Table S13). This indicates that >75% (228/306) of *A. thaliana* flowering-time genes have homologs in *S. curtisii,* similar to the

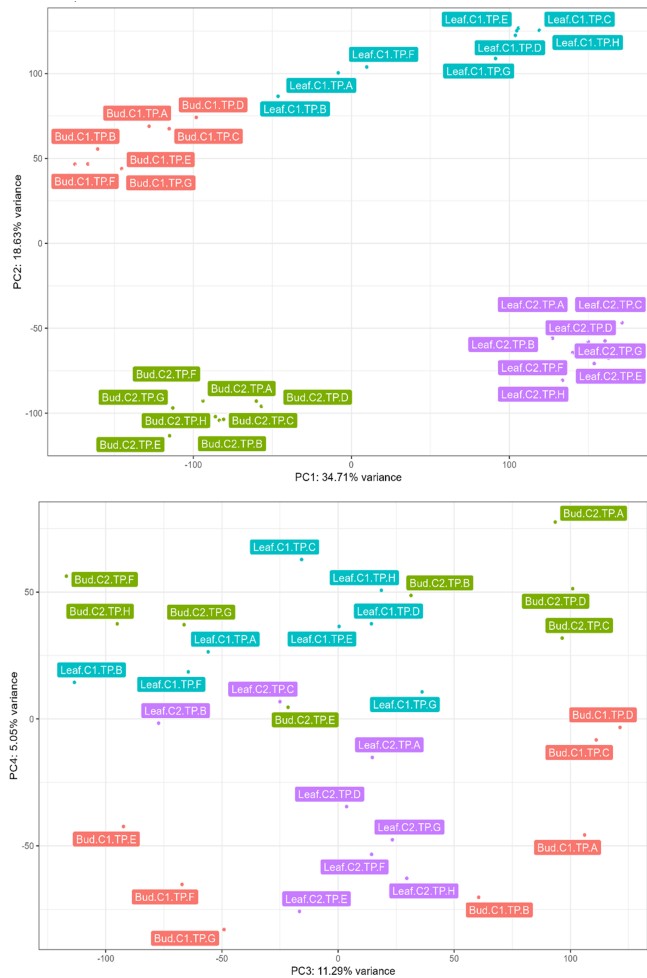

**Figure 1** **Principal component analysis conducted on normalized expression data of leaf and bud samples from two *Shorea curtisii* individuals collected at eight time points during a general flowering year.** (A) Score plot of the first principal component (PC1; *x*-axis) against the second principal component (PC2; *y*-axis). (B) Score plot of the third principal component (PC3; *x*-axis) against the fourth principal component (PC4; *y*-axis).

findings in a closely related species, *S. beccariana* (*Kobayashi et al., 2013*). Out of the total *A. thaliana* flowering-time homologs, 103 were differentially expressed throughout the study period (Table S13).

There are 274 flowering-time homologs associated with the photoperiod and circadian clock pathway (Table 1 and Table S13). Photoperiod-associated homologs accounted for over half (15/26 DEUs) of flowering-time DEUs in leaf, while only one third (37/96 DEUs) in bud were from this pathway (Table S13). The number of homologs related to the circadian clock and photoperiod pathway in this study is consistent with our earlier study in *S. curtisii* leaves despite using fewer time points in the previous study (*Suhaimi et al., 2023*). We also identified 99 homologs belonging to the vernalization pathway, 34 homologs from the autonomous pathway, and 16 ambient temperature-responsive
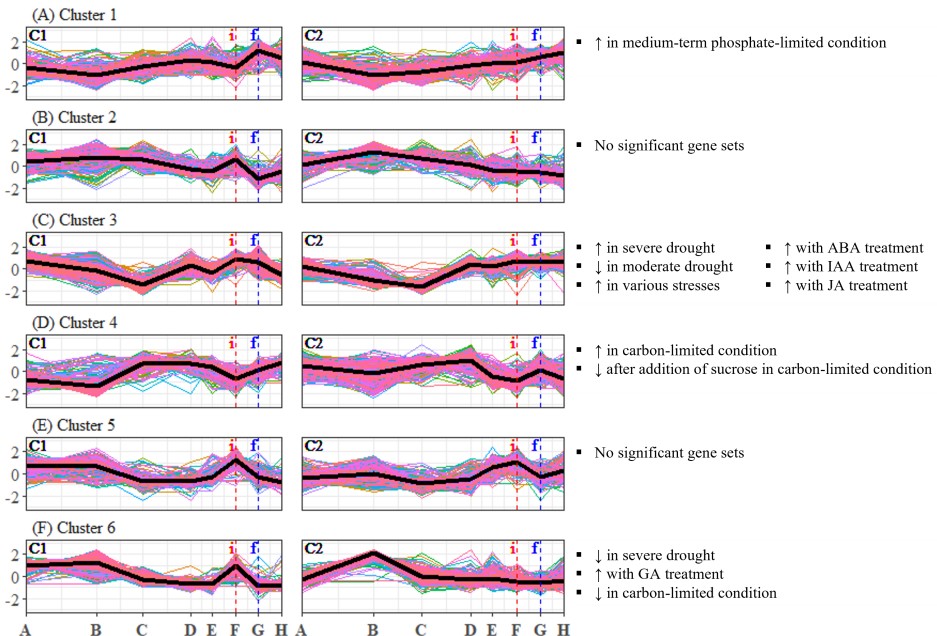

**Figure 2** **(A–F) Expression profiles of differentially expressed unigene clusters in *Shorea curtisii* leaf and corresponding significantly enriched *Arabidopsis thaliana* gene sets.** The *y*-axis represents the expression of unigenes, scaled to a mean of 0 and a standard deviation of 1 across the samples. The *x*-axis represents the time points. The black lines represent the mean expression pattern of each cluster. Dashed vertical lines indicate the time when inflorescence (i) and flowering stages (f) were observed. *Arabidopsis thaliana* gene sets that have significant number of overlapping unigenes with each differentially expressed unigene cluster are shown on the right side of each profile (refer to Table S22 for detailed result of the enrichment analysis). (i): Inflorescence stage; (f): Flowering stage; C1: Individual 1, C2: Individual 2, ↑ : Up-regulated genes, ↓ : Downregulated genes.

homologs in the transcriptome. Of the homologs from the vernalization pathway, eight were differentially expressed only in bud, while two homologs of *VERNALIZATION 5* (*VRN5*) were differentially regulated in both leaf and bud (Table 1). Only two homologs from the autonomous pathway were differentially expressed in bud (Table 1). Of the homologs associated with ambient temperature, two homologs of *RELATED TO ABI3/VP1 2* (*RAV2*) were differentially expressed in both leaf and bud (Table S13). Additionally, we identified 41 homologs of integrators in this study, ten of which were differentially expressed only in bud. Two homologs of *FT*, namely *ScFT1* and *ScFT2*, were differentially regulated in both tissues (Table S13). The expression of these *FT* homologs is congruent with qRT-PCR data from a previous study (*Yeoh et al., 2017*; Fig. S7), thus, validating the RNA-Seq results.

## Characterization of the differentially expressed unigenes through *A. thaliana* gene sets enrichment analyses

Only 2.7% of the total DEUs (122/4,500) are homologs of *A. thaliana* flowering-time genes (Tables S11–13), which suggests that most of them have functions that are not directly related to known flowering mechanisms in the model species. To characterize these DEUs, enrichment analyses were performed on 35 gene sets of *A. thaliana* (Table 2) and the DEUs

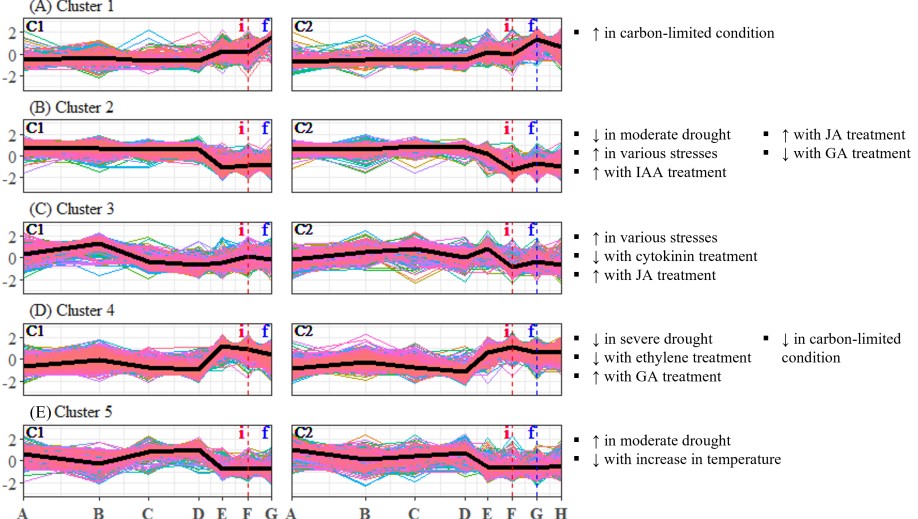

**Figure 3** **(A–E) Expression profiles of differentially expressed unigene clusters in *Shorea curtisii* bud and corresponding significantly enriched *Arabidopsis thaliana* gene sets.** The *y*-axis represents the expression of unigenes, scaled to a mean of 0 and a standard deviation of 1 across the samples. The *x*-axis represents the time points. The black lines represent the mean expression pattern of each cluster. Dashed vertical lines indicate the time when inflorescence (i) and flowering stages (f) were observed. *Arabidopsis thaliana* gene sets that have significant number of overlapping unigenes with each differentially expressed unigene cluster are shown on the right side of each profile (see Table S29 for detailed results of the enrichment analysis). (i): Inflorescence stage; (f): Flowering stage; C1: Individual 1, C2: Individual 2, ↑ : Upregulated genes, ↓ : Downregulated genes.

**Table 1** **Summary of *Arabidopsis thaliana* flowering-time homologs and differentially expressed unigenes (DEUs) identified in the leaf and bud transcriptomes of *Shorea curtisii*.**

| Floral regulatory pathway | No. of *A. thaliana* genes | No. of *S. curtisii* homologs | No. of DEUs | |
|---|---|---|---|---|
| | | | Leaf | Bud |
| General | 117 | 402 (100) | 2 (2) | 22 (12) |
| Photoperiod and circadian clock | 103 | 274 (70) | 15 (9) | 37 (20) |
| Hormones | 28 | 62 (21) | 3 (3) | 5 (4) |
| Vernalization | 28 | 99 (21) | 2 (1) | 10 (4) |
| Aging | 22 | 23 (6) | 0 (0) | 8 (4) |
| Floral meristem identity | 9 | 26 (6) | 0 (0) | 7 (4) |
| Sugar | 9 | 46 (9) | 2 (1) | 9 (3) |
| Integrator | 8 | 41 (7) | 2 (2) | 12 (4) |
| Ambient temperature | 7 | 16 (5) | 2 (1) | 2 (1) |
| Autonomous | 7 | 34 (7) | 0 (0) | 2 (1) |

**Notes.**
Number in parentheses indicates the number of non-redundant *A. thaliana* genes that corresponds to *S. curtisii* homologs.

separately for leaf and bud samples. We found 25 *A. thaliana* gene sets to be significantly enriched, that is, these gene sets share significant number of overlapping genes with the

DEUs for both *S. curtisii* leaf and bud (Table 2; Tables S14 and S15). These gene sets include responses to drought and temperature, as well as changes in level of hormones such as jasmonic acid (JA) and gibberellin (GA). We assumed that the trees experienced similar conditions as the *A. thaliana* from which the gene sets were obtained when the tests were significant.

## Expression profiles of the enriched gene sets

To analyze the expression changes of the enriched gene sets across the study period, enrichment analysis was conducted on the gene sets that were found to be significant (Table 2) and each DEU cluster in leaf and bud of *S. curtisii* (Figs. 2 and 3). We assumed that the expression pattern of gene sets enriched in a particular DEU cluster is represented by that of the cluster. Overall, 12 gene sets were enriched in at least one DEU cluster in leaf (Fig. 2, Table S23) while 13 gene sets were overrepresented in at least one DEU cluster in bud (Fig. 3 and Table S30). Notably, a number of gene sets related to nutrient levels, biotic and abiotic stresses, and hormonal treatments were enriched in DEU clusters that showed similar expression in both trees (Figs. 2 and 3).

Our analysis of DEU clusters in leaf has found that the homologs of *A. thaliana* genes that are upregulated under medium-term phosphate-limited condition were enriched in DEU cluster 1 (Fig. 2A). The expression profile of this cluster indicates a decrease in phosphorus level between TP-B and TP-D, followed by another decrease prior to flowering. DEU cluster 3 in leaf was overrepresented with homologs of genes commonly upregulated in response to biotic and abiotic stresses, as well as genes upregulated with hormonal treatment, specifically abscisic acid (ABA), indole-3-acetic acid (IAA, or commonly known as auxin), and JA (Fig. 2C). These phytohormones are also involved in plant stress response (*Yang et al., 2019*). Furthermore, GO terms related to stress response such as "response to reactive oxygen species" and "MAPK cascade", were enriched in this cluster (Table S34). The expression of the DEUs was elevated from TP-D to TP-F (Fig. 2C), suggesting that the plants were under stress during this period. DEU cluster 4 in leaf was enriched in genes that are upregulated under carbon-limited condition and downregulated with addition of sucrose (Fig. 2D). The expression profile of this cluster suggests that the concentration of carbon in the leaf was decreasing at the onset of flowering, whilst sucrose level was increasing prior to floral induction, specifically at TP-E in C1 and TP-D in C2. Additionally, the KEGG pathway analysis identified the "carbon metabolism" pathway as being enriched in this cluster (Table S34). Finally, DEU cluster 6 in leaf was overrepresented with gene sets responding to drought and changes in the levels of GA and carbon (Fig. 2F). However, the expression patterns of the DEUs in the cluster varied between the two individuals, implying that the response observed was specific to each individual. As a result, the enriched gene sets in the cluster were not further analyzed.

Examination of DEU clusters in bud identified that the homologs of *A. thaliana* genes upregulated in response to carbon-limited condition were enriched in DEU cluster 1 (Fig. 3A). The expression pattern of the cluster suggests that the level of carbon in bud was depleted as flowering approached (Fig. 3A), which is consistent with the results of enrichment tests in leaf (Fig. 2D). GO terms related to catabolism such as "polysaccharide

**Table 2** Overrepresentation of *Arabidopsis thaliana* gene sets in the differentially expressed unigenes of *Shorea curtisii*.

| No. | Description of *A. thaliana* gene set | Leaf | Bud |
|-----|---------------------------------------|------|-----|
| 1. | Upregulated under severe drought | 4.2E–11 | NS |
| 2. | Downregulated under severe drought | 1.3E–05 | 7.7E–15 |
| 3. | Upregulated under prolonged moderate drought | 8.0E–06 | 7.7E–15 |
| 4. | Downregulated under prolonged moderate drought | 7.7E–15 | 7.7E–15 |
| 5. | Commonly upregulated under 41 abiotic and biotic stress conditions | 7.7E–15 | 7.7E–15 |
| 6. | Upregulated with temperature increase | 3.3E–12 | 1.3E–06 |
| 7. | Downregulated with temperature increase | 7.7E–15 | 7.7E–15 |
| 8. | Upregulated with temperature decrease | NS | NS |
| 9. | Downregulated with temperature decrease | 1.8E–04 | NS |
| 10. | Upregulated with abscisic acid treatment | 7.7E–15 | 7.7E–15 |
| 11. | Downregulated with abscisic acid treatment | NS | 1.1E–04 |
| 12. | Upregulated with brassinosteroid treatment | 2.3E–12 | 7.7E–15 |
| 13. | Downregulated with brassinosteroid treatment | NS | NS |
| 14. | Upregulated with cytokinin treatment | NS | NS |
| 15. | Downregulated with cytokinin treatment | 1.3E–08 | 5.0E–07 |
| 16. | Upregulated with ethylene treatment | NS | 5.3E–06 |
| 17. | Downregulated with ethylene treatment | 3.6E–06 | 7.7E–15 |
| 18. | Upregulated with indole-3-acetic treatment | 7.7E–15 | 7.7E–15 |
| 19. | Downregulated with indole-3-acetic treatment | NS | 7.4E–06 |
| 20. | Upregulated with jasmonic acid treatment | 7.7E–15 | 7.7E–15 |
| 21. | Downregulated with jasmonic acid treatment | 1.1E–12 | 7.7E–15 |
| 22. | Upregulated with gibberellin in young flower buds | 7.7E–15 | 7.7E–15 |
| 23. | Downregulated with gibberellin in young flower buds | 1.2E–11 | 7.7E–15 |
| 24. | Upregulated under carbon-limited condition | 7.7E–15 | 2.5E–09 |
| 25. | Downregulated under carbon-limited condition | 7.7E–15 | 7.7E–15 |
| 26. | Upregulated after the addition of sucrose under carbon-limited condition | 7.7E–15 | 7.7E–15 |
| 27. | Downregulated after the addition of sucrose under carbon-limited condition | 1.7E–11 | NS |
| 28. | Upregulated under long-term phosphate-limited condition | 7.7E–15 | 7.7E–15 |
| 29. | Downregulated under long-term phosphate-limited condition | NS | NS |
| 30. | Upregulated under medium-term phosphate-limited condition | 1.6E–08 | 6.2E–04 |
| 31. | Downregulated under medium-term phosphate-limited condition | NS | NS |
| 32. | Upregulated under shot-term phosphate-limited condition | 2.0E–10 | NS |
| 33. | Downregulated under short-term phosphate-limited condition | NS | NS |
| 34. | Upregulated under nitrogen-limited condition | 1.1E–09 | 1.4E–08 |
| 35. | Downregulated under nitrogen-limited condition | NS | 2.5E–08 |

**Notes.**
Fisher's exact test *P*-value cut-off: 1E–03 after Bonferroni correction was used.
NS indicates not significant.

catabolic process", "cellular amide catabolic process", and "allantoin catabolic process", were also found to be overrepresented in the cluster (Table S35). This aligns with previous research in *A. thaliana* which demonstrated that carbon starvation leads to an upregulation of genes involved in catabolism (*Osuna et al., 2007*). Among the DEU clusters in bud, clusters 2 and 4 showed inverse expression patterns in both individuals (Figs. 3B and 3D). Cluster 2 was enriched with homologs of genes that are downregulated under moderate drought, while cluster 4 was not. The expression profile of DEU cluster 2 suggests that the *Shorea* individuals were experiencing mild drought between TP-D and TP-F. Additionally, both clusters 2 and 4 were enriched with homologs of genes that respond to elevated level of GA. The expression profiles of these two clusters indicate an increase in GA level between TP-D and TP-F. Both clusters were also enriched with GO terms related to stress, signaling, growth, and floral regulation, such as "response to reactive oxygen species", "vasculature development", "MAPK cascade", and "signal transduction" in cluster 2, as well as "response to hypoxia", "response to light stimulus", "microtubule cytoskeleton organization", and "cytokinesis" in cluster 4 (Table S35). Our PCA also revealed that the majority of top contributing unigenes to PC3 belong to these two clusters (Table S10), highlighting the importance of the unigenes in these clusters in the transition from vegetative to reproductive stages. In DEU cluster 3, we found homologs of *A. thaliana* genes that are typically upregulated in response to diverse abiotic and biotic stresses, as well as genes that are affected by cytokinin and JA treatment (Fig. 3C). However, this cluster showed inconsistent expression profiles between the two *S. curtisii* individuals (Fig. 3C), hence, the enriched gene sets in this cluster were not examined further. Lastly, DEU cluster 5 in bud was overrepresented with homologs of genes upregulated under moderate drought and downregulated with increasing temperature (Fig. 3E). The expression profile of this cluster indicates an increase in temperature between TP-D and TP-E (Fig. 3E), which is in line with the enrichment of relevant GO terms such as "response to stress" and "response to stimulus" (Table S35).

## Association between meteorological data and transcriptome profiles

Prolonged moderate drought and drop in temperature have been hypothesized as proximate cues for GF (*Ashton, Givnish & Appanah, 1988*; *Brearley et al., 2007*; *Chen et al., 2018*; *Yeoh et al., 2017*). Therefore, we examined meteorological data to look for these signals prior to flowering. These data were then compared to the expression patterns of DEU clusters that were significantly enriched with *A. thaliana* gene sets related to temperature or drought (Figs. 2 and 3) to infer the regulatory pathways that respond to these climatic cues.

The daily mean temperature at the nearest meteorological station situated 11 km from the study site was 26.8 °C, while the total annual precipitation at the nearest hydrological station 6 km from the study site was 2,272 mm. The lowest minimum daily temperature (18.3 °C) was recorded in early February around TP-D (Fig. 4B). We also calculated 30-day cumulative rainfall to identify drought periods, which are defined as 30-day cumulative rainfall of below 40 mm (*Sakai et al., 2006*). The cumulative rainfall data showed a period of drought beginning before TP-D and lasting until TP-E (Fig. 4C). Therefore, the expression

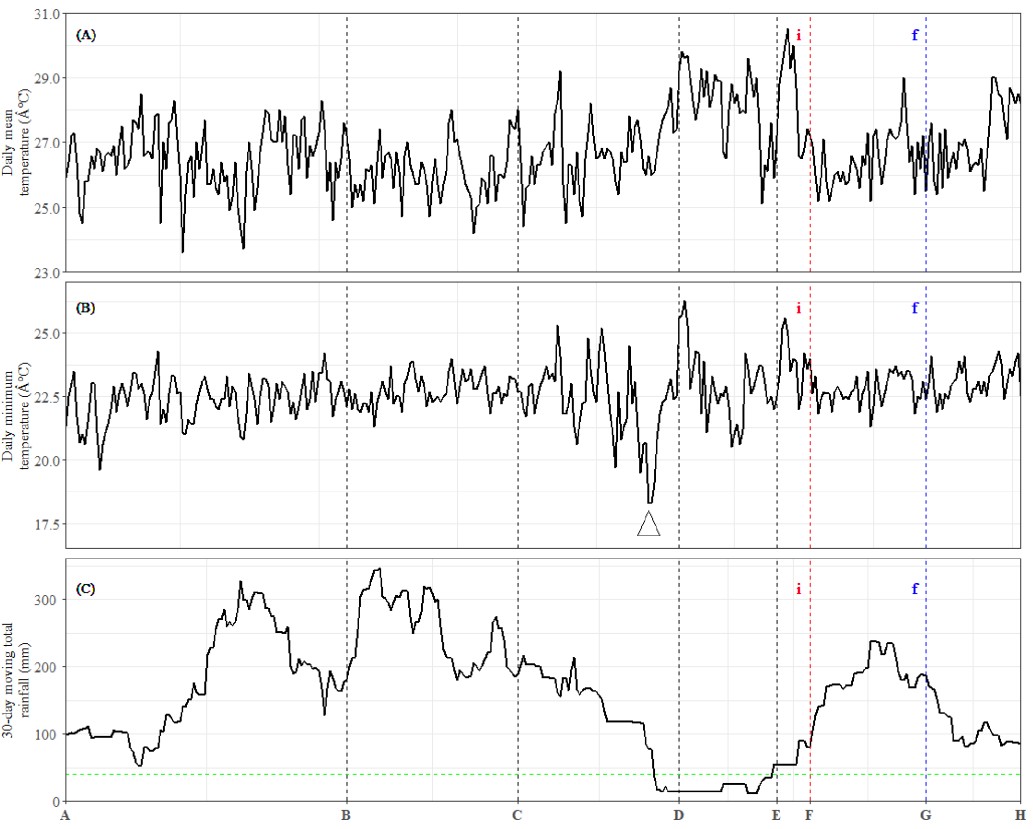

**Figure 4** **Meteorological data and occurrences of general flowering from July 2013 to June 2014.** (A) Daily mean temperature. (B) Daily minimum temperature. The arrow indicates the sharpest drop in daily minimum temperature during the study period. (C) Total rainfall over the preceding 30-day period. The dashed horizontal line represents the 40-mm drought threshold. Dashed vertical lines indicate the time points (*x*-axis). (i): Inflorescence stage; (f): Flowering stage.

patterns of leaf DEU cluster 3 (Fig. 2C) as well as bud DEU clusters 2, 4, and 5 (Figs. 3B, 3D and 3E) were examined.

Among these DEU clusters, only DEU cluster 5 in bud was enriched with homologs of genes downregulated with increased temperature (Fig. 3E) and its expression pattern implies that the plants were responding to rising temperature around TP-D. This pattern concurs with the daily mean temperature data, which showed a slight increase within the same period (Fig. 4A). Although the increase in temperature was subtle, it was not unexpected for the plants to be able to capture the slight change in temperature stimuli as it has been suggested that trees in tropical regions are more sensitive to change in their surroundings (*Singh & Kushwaha, 2016*). Of the DEU clusters overrepresented with homologs of genes responding to drought, only DEU cluster 3 in leaf (Fig. 2C) and cluster 2 in bud (Fig. 3B) displayed expression patterns consistent with the drought period (Fig. 4C). This indicates that the trees were experiencing a moderate drought between TP-D and TP-E. The expression pattern of DEU cluster 5 in bud (Fig. 3E), on the other hand, did not match the cumulative rainfall data.

In essence, we assembled a transcriptome with majority of the unigenes annotated. The examination of these annotated unigenes, along with the differential expression analysis, identified homologs of flowering-time genes associated with various pathways, such as photoperiod, circadian clock, and ambient temperature responses. Subsequent enrichment analyses and the expression profiles of DEU clusters suggested that the plants responded to factors such as phosphate and carbon availability, and hormonal dynamics. Additionally, the examination of the meteorological data in relation to the DEU clusters indicated that the trees exhibited changes in transcriptional expression in response to rising temperatures and a brief spell of drought.

## DISCUSSION

Our study found high transcriptional variation between leaf and bud of *S. curtisii* during GF, including the expression of *A. thaliana* homologs involved in flowering. Despite the inter-individual differences in the expression profiles, we were able to identify and characterize transcriptional profiles of unigenes that are differentially expressed in the vegetative and reproductive stages of the trees. We interpreted the outcome of our transcriptome analysis along with available meteorological data and proposed a preliminary framework of floral initiation and development in this GF dipterocarp species.

### Proximate environmental cues for general flowering

The increasing rate of climate change, characterized by higher temperature fluctuations (*IPCC, 2022*) and more frequent and severe droughts (*Dai, 2013*; *Trenberth et al., 2014*), among others, is expected to have a disproportionate impact on the phenology of plants and animals in tropical regions (*Santer et al., 2018*). The fitness and abundance of tropical communities, including GF dipterocarp species, can be adversely affected by a slight temperature increase (*Lister & Garcia, 2018*; *Numata et al., 2022*). As these plant species is a valuable source of sustenance for a variety of seed and fruit predators including wild pigs and rodents (*Ickes, 2001*; *Miura, Yasuda & Ratnam, 1997*), alteration in their phenology could have potential cascading effects on the tropical forest communities.

Long term phenological records of tropical rainforests in the Malay Peninsula have shown that flowering events are always preceded by a drop in daily minimum temperature after a brief period of drought (*Ashton, Givnish & Appanah, 1988*; *Sakai et al., 2006*). In fact, mathematical models that take into account the synergistic effects of both drought and a drop in daily minimum temperature have been found to be reliable at predicting GF occurrences (*Chen et al., 2018*; *Yeoh et al., 2017*). The cumulative rainfall data (Fig. 4C), along with the expression profile of DEU clusters that were enriched with homologs of drought-responsive genes (Figs. 2C and 3B) indicate that the *S. curtisii* in our study experienced water stress due to a moderate drought lasting from before TP-D until TP-E. The upregulation of *ScFT1* (Unigene: TRINITY_DN6921_c0_g2_i2) in leaf shortly after drought began (TP-D; Table S11) and the upregulation of *ScFT1* in bud post-drought (TP-E; Table S12) further corroborated the hypothesis that drought is one of the floral triggers in dipterocarps (*Kobayashi et al., 2013*; *Sakai et al., 2006*; *Yeoh et al., 2017*). Moreover, the observed expression changes in homologs associated with elevated levels of ABA, IAA, and

JA hormones (Fig. 2C) that coincides with the occurrence of drought also supported the aforementioned hypothesis. Roles of these hormones in the floral regulation and signaling pathways of plants responding to stress, including drought, have been established in many other species such as *A. thaliana*, rice and *Citrus* spp. (reviewed in *Singh & Laxmi, 2015*; *Yang et al., 2019*). Among these hormones, ABA is particularly important in the hormonal response to drought stress (*Gupta, Rico-Medina & Caño Delgado, 2020*) as it regulates stomatal opening to reduce water loss (*Okamoto et al., 2013*). Interestingly, ABA has also been reported to function in drought-induced flowering in *A. thaliana* (*Riboni et al., 2016*) and *Citrus* spp. (*Khan et al., 2022*; *Li et al., 2017*). Auxin (IAA) plays a role in modulating leaf water uptake and regulating the expression of antioxidant enzymes to detoxify reactive oxygen species (*Shi et al., 2014*). Similarly, JA contributes to abiotic stress responses by increasing resistance to oxidative stress (*Wu et al., 2012*) and maintaining downstream drought-responsive pathways such as those related to modulation of shoot biomass and photosynthetic rate (*Wang et al., 2021*).

Despite a sharp drop in daily minimum temperature observed prior to TP-D (Fig. 4B), we did not find any gene set associated with differential regulation in response to decrease in temperature to be overrepresented in any of the DEU clusters (Figs. 2 and 3). However, several studies have shown that temperature drops always precede GF (*Ashton, Givnish & Appanah, 1988*; *Numata et al., 2003*; *Yasuda et al., 1999*), and that the occurrence of drought alone does not always result in GF (*Chen et al., 2018*; *Yeoh et al., 2017*). Therefore, it is possible that *S. curtisii* employs different temperature-responsive mechanisms than *A. thaliana*. This scenario is not unlikely, as similar divergence has been reported in other floral regulatory pathways. For example, *FLOWERING LOCUS C* (*FLC*) encodes a key floral repressor in the vernalization pathway of *A. thaliana*, grasses have developed vernalization response that is completely independent of FLC, relying instead on four central genes: *VERNALIZATION 1–3* (*VRN1–3*) and *VEGETATIVE TO REPRODUCTIVE TRANSITION 2* (*VRT2*) (reviewed in *Trevaskis et al., 2007*). Further studies using transcriptome profiles of GF dipterocarp trees treated with low temperature and drought under controlled environments could help to verify the potential of low temperature as a flowering cue and further prove that a short period of drought can induce flowering in the species. However, this would be a practically challenging and costly endeavor for tall tropical trees that flower irregularly.

Additionally, several homologs of flowering-time genes were found to be differentially expressed after the short dry spell (Figs. 2 and 3; Tables S11–13). Given the accumulating evidence of organisms and populations adapting to climate change through phenotypic plasticity and genotypic evolution (reviewed in *Peñuelas et al., 2013*), further research focusing on these homologs could help us understand how these rapid environmental changes will impact the reproductive phenology of GF dipterocarp species and how this phenomenon will evolve in the future.

### Resource dynamics in *Shorea* during flowering time

Accumulated nutrients have been proposed as a limiting factor in GF, which could explain why some mature trees do not flower during GF (*Ichie et al., 2013*; *Kelly & Sork, 2002*).

In the present study, we found molecular evidence showing changes in the expression levels of homologs associated with a decrease in phosphorus level prior to flowering (Fig. 2A). This finding is consistent with a previous report that investigated mineral nutrient storage dynamics prior to mast reproduction in another dipterocarp species, *Dryobalapnos aromatica* (*Ichie & Nakagawa, 2013*). Although we identified homologs of *A. thaliana* genes responsive to nitrogen to be overrepresented in the DEUs of *S. curtisii* (Table 2), these unigenes did not show any specific expression pattern (Figs. 2 and 3). In addition to phosphorus and nitrogen, carbohydrates have been demonstrated to be utilized for floral initiation and subsequent development of floral organs (*Peng & Iwahori, 1994*). The results of our cluster analysis indicate changes in the expression level of unigenes correlated with an increase in sucrose level in leaves during the brief period of drought (Fig. 2D). An elevated level of sucrose in leaves during drought have been reported in other trees such as East Asian white birch (*Bhusal et al., 2021*) and tropical starfruit (*Pingping, Chubin & Biyan, 2017*). Furthermore, the cluster analysis identified changes in expression levels of homologs correlated with a reduction in carbon concentrations in bud prior to flowering time (Fig. 3A). Previous studies have suggested that flower bud act as carbohydrate 'utilizing sinks', metabolizing rather than storing carbohydrate (*Eshghi et al., 2007*; *Ho, 1988*).

In the current study, we inferred the fluctuation of nutrient levels based on associated or correlated transcript expressions. To gain a more comprehensive understanding of the roles and dynamics of plant resources during flowering time, future studies should consider integrating nutritional content information such as nitrogen, phosphorus and carbon concentrations. The collection of sufficient tissues from carbohydrate sinks such as stems and trunks (*Kozlowski, 1992*) without damaging the branches was not feasible in this study due to the need for frequent repeated sampling to capture the floral initiation time points of these trees with unpredictable flowering intervals. Furthermore, canopy access for tall tropical trees has thus far been limited, hence observations and sample collections in our study were done by climbing approximately 40 m above the ground canopy. It is also important to acknowledge the low statistical power of this study due to the limited number of biological replicates and the absence of technical replicates. In addition, due to logistical constraints, rainfall measurements in the current study were obtained from the nearest hydrological station located adjacent to the forest reserve, rather than being recorded at the specific location of each individual tree. Therefore, to address these limitations, it would be essential to employ improved monitoring and sampling techniques *e.g.*, by utilizing technology such as unmanned aerial vehicles (drones) for sample collection, to facilitate future research. Notwithstanding these challenges and limitations, we were able to conduct the present study on two individual trees and obtain clear and consistent results.

Based on the findings in the current study and earlier reports (*Chen et al., 2018*; *Ichie et al., 2013*; *Kobayashi et al., 2013*; *Yang et al., 2019*; *Yeoh et al., 2017*), we propose a preliminary framework that depicts the interactions between external environmental cues, phytohormones, nutrient resources and expression of florigen, *FT* in *Shorea* (Fig. S9) during GF. According to this framework, flowering signals such as drought (and potentially, drops in daily minimum temperature) accumulate over a period of 2–3 months prior to floral initiation. During this period, the levels of stress-associated phytohormones, including

IAA, ABA, and JA, increase, leading to the upregulation of the key flowering gene, *FT* in the leaf. Subsequently, *FT* expression in the bud is elevated, thus initiating the transition to the reproductive stage. Further research is needed to develop and complete this proposed framework and determine the extent of its applicability to other GF dipterocarp species.

## CONCLUSIONS

To date, this is the first genome-wide transcriptome sequencing study conducted concurrently on both leaf and bud of a dipterocarp species over a flowering season. The characterization of *S. curtisii* transcriptome in this study has led to the identification of numerous unigenes that are homologous to *A. thaliana* flowering-time genes, including unigenes that were differentially expressed during the flowering season. We have also identified unigenes that are important during the switch from vegetative to reproductive stages but are not known to be involved in the *A. thaliana* flowering mechanism, hence warrant further investigation. Our ecological transcriptome approach, which involved cluster analysis of DEUs in *S. curtisii* and enrichment analysis with *A. thaliana* gene sets and climatic data, suggests that the trees perceived proximate flowering cues, such as drought, approximately three months prior to floral initiation. This was followed by changes in expression of genes associated with hormone levels, as well as phosphorus and carbohydrate contents, as the trees entered reproductive stages. The outcomes of this study provide insight into the molecular and physiological mechanisms underlying floral regulation in a GF dipterocarp species and highlight the need for further research in this area to fully elucidate these complex processes.

## ACKNOWLEDGEMENTS

The authors would also like to thank B. Yasri and P. Ramli for their assistance in the fieldwork, and Dr. KKS Ng for his assistance in RNA extraction. We would like to thank the reviewers for their feedback and comments, which helped to improve our manuscript. This research was supported in part through computational resources provided by the Data Intensive Computing Centre, Universiti Malaya.

### Funding

This research was funded by grants from MoE-HIR (UM.C/625/1/HIR/MOE/SCI/18) awarded to Ching Ching Ng and Suat Hui Yeoh and BKP (BKS074-2017) awarded to Suat Hui Yeoh as well as Environment Research and Technology Development Fund (Grant No. RFd-1101) from The Ministry of the Environment, Japan, KAKENHI grant (26251042), and Grant-in-Aid for Transformative Research Areas (23A401) awarded to Akiko Satake. Ahmad Husaini Suhaimi was supported by the Universiti Malaya's SLAB fellowship. The funders had no role in study design, data collection and analysis, decision to publish, or preparation of the manuscript.

## Grant Disclosures

The following grant information was disclosed by the authors:
MoE-HIR: UM.C/625/1/HIR/MOE/SCI/18, BKS074-2017.
Environment Research and Technology Development Fund: RFd-1101.
KAKENHI: 26251042.
Grant-in-Aid for Transformative Research Areas: 23A401.
Universiti Malaya's SLAB fellowship.

## Competing Interests

The authors declare there are no competing interests.

## Author Contributions

- Ahmad Husaini Suhaimi performed the experiments, analyzed the data, prepared figures and/or tables, authored or reviewed drafts of the article, and approved the final draft.
- Masaki J. Kobayashi analyzed the data, authored or reviewed drafts of the article, and approved the final draft.
- Akiko Satake conceived and designed the experiments, authored or reviewed drafts of the article, and approved the final draft.
- Ching Ching Ng analyzed the data, authored or reviewed drafts of the article, and approved the final draft.
- Soon Leong Lee performed the experiments, authored or reviewed drafts of the article, and approved the final draft.
- Norwati Muhammad performed the experiments, authored or reviewed drafts of the article, and approved the final draft.
- Shinya Numata performed the experiments, authored or reviewed drafts of the article, and approved the final draft.
- Tatsuya Otani performed the experiments, authored or reviewed drafts of the article, and approved the final draft.
- Toshiaki Kondo performed the experiments, authored or reviewed drafts of the article, and approved the final draft.
- Naoki Tani conceived and designed the experiments, analyzed the data, authored or reviewed drafts of the article, and approved the final draft.
- Suat Hui Yeoh conceived and designed the experiments, analyzed the data, authored or reviewed drafts of the article, and approved the final draft.

## Data Availability

The transcriptome assembly is available at BioProject: PRJNA768952; GKBV00000000.
The R code for the different bioinformatic analyses conducted in this study is available at Protocols.io: https://www.protocols.io/workspaces/mp2lab:
https://dx.doi.org/10.17504/protocols.io.b2ipqcdn;
https://dx.doi.org/10.17504/protocols.io.b2iqqcdw;
https://dx.doi.org/10.17504/protocols.io.rm7vzx82rgx1/v1.

## Supplemental Information

Supplemental information for this article can be found online at http://dx.doi.org/10.7717/peerj.16368#supplemental-information.

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
