# Peer review of "An ecological transcriptome approach to capture the molecular and physiological mechanisms of mass flowering in Shorea curtisii"

_PeerJ, doi:10.7717/peerj.16368_

## Round 0.1 · original submission · Major Revisions

We have received two reviews for your manuscript. Both reviewers found merits in your study, but raised a number of very important concerns regarding the methods and analyses performed. Reviewers also identified problems with the general context of the introduction, the objectives of the study and some parts of the discussion.

Given these problems, I am leaving the possibility to resubmit open, but only if the authors feel they can address the comments provided (especially those concerning the experimental design and analyses) convincingly.

Reviewer 1 ·

Basic reporting

The manuscript is well-written throughout and the language is clear. The Introduction provides ample background information about general flowering (GF) in Shorea and dipterocarps, includes some of the hypothesized cues and regulators of GF, and the stated rationale for conducting this study.

I could not find anywhere in the manuscript where the raw RNA-seq data was deposited. It would also be helpful toward reproducibility and transparency if the R code for the different analyses was shared in a free repository such as Github or Zenodo.

Experimental design

This study sampled multiple tissues (leaf and bud) across multiple time points. The aim of the work is to identify differentially expressed transcripts across the time series that leads to GF.

I have major questions about how the DE analyses were performed. It was not clear from the manuscript text what comparisons were being made among the different time points. This needs to be clarified in a revised manuscript and examples should be provided.
A better approach to analyze time-series RNA-seq data would be to treat the first time point as a control and compare each subsequent time points to this control. For example, TP-A was the first sample and could serve as the control. Then, each other time point (TP-B to TP-H) would be compared to TP-A (TP-A vs. TP-B; TP-A vs. TP-C; TP-A vs. TP-D; etc.). This would identify differentially expressed genes for each time point individually. I think this approach would provide more meaningful results and would be a more statistically sound approach.

Validity of the findings

It is clear from the Discussion that sampling additional individuals to increase the number of biological replicates would be a challenge. I think this is valid. However, the authors should clearly state somewhere near the beginning of the manuscript (maybe in the Materials & Methods) why only two individual trees were sampled. Some explanation or justification should be included. Given the unique challenges of sampling from the canopy of these trees, I think the research is still very valid.

In several parts of the Discussion, the language should be softened and revised because the claims being made were not actually tested in the current study. For instance, on Line 408 the authors state that "elevation in the levels of ABA, IAA, and JA hormones that coincides with the occurrence of drought". This reads as if these hormone levels were directly measured in the current study, but the only measurement was the expression levels of transcripts associated with the production of these hormones. Similarly this happens in other places on Lines 447,455, and 458 regarding sucrose, carbon, and phosphorus levels. These claims need to be revised to clarify that only transcript expression associated or correlated with these levels were measured.

Additional comments

Line 68 - Please define ENSO.

·

Basic reporting

In this paper Suhaimi et al. used a transcriptomic approach to understand Mass Flowering from a molecular and physiological perspective. In this paper they give a fresh approach to ecology and in particular mass flowering by combining gene differential expression data with rainfall data. They found differentially expressed genes related to stress, nutritional levels and hormonal changes. Overall they proposed a model combining this all this information to explain mass flowering in Shorea.

Overall I think the idea of this paper is really neat and I see the field of ecology combining techniques like to answer very interesting questions related to physiology, evolution among other and in non-model organisms. I also would like to applaud the authors for takin the time of such to do this research since it seems difficult to reach the canopy of this trees to do the tissue collecting.

My main concerns about the paper have to do with the lack of a well structure and more robust introduction to the topic and very vague objectives. I understand authors would like to answer the question of General Flowering in a dipterocarp forest, but that is a much bigger question to the answer authors actually have here. So I recommend them to focus on the specific question they can answer with these results and briefly touch on the general flowering topic in their discussion as future endeavors. In that order of ideas, I think authors really need to refocus the whole introduction since they mentioned too many disparate topics. The molecular basis of flowering in trees and what we know in physiological, hormonal and molecular basis should be mentioned here, only drawing examples on Arabidopsis if absolutely necessary and for informative/comparative purposes only. Additionally the introduction lack very basic information about the relationship between rain fall and flowering in trees.

Experimental design

The experimental design is not robust enough, they just use two replicates, and statistically this is very poor sampling even with the sampling effort they have to go through. Maybe less time points and more replicates will have been a better choice here, however it is hard to know because the reader does not have a good background in the introduction about the developmental times they choose to do this sampling. For the time points the authors selected it is not clear what they meant by A-H ( authors cited a previous paper but this should be explain here), they do not give an explanation about the developmental points they chose and why. They should have picked less points and sequence more trees or replicates. I also think they should have used more than three replicates per time point in each individual to make their study more robust. Now, the labeling of the developmental points gets even more confusing in the results section. Now I think that I maybe misinterpreting the results because the methods are not clearly written.Are A-D vegetative samples or leaf tissue? so basically replicates within in a tree? Im unsure (see results lines 232-243).
The use of Arabidopsis as a comparative system here, I will say is a bit confusing, there are already some many tools (some of them the authors already used), to make you study more comparatively broad in terms of the tools you are using.
Rainfall measurements should have been taken from the specific location of each tree.

Validity of the findings

The gene expression variance between tissues and and replicated is really high as expected based on the experimental design. Some of the figures that are in supplementary materials should be included as main figures in the paper (like the pea analysis of the variance between TP) and figures 1, 2, 5 are unnecessary.
Figure 3 (What do authors mean by significantly enriched A. thaliana gene sets? I thought authors have only used A. thaliana no annotate homologs? which I think it is not the best way to have annotate this. also the following statement is really hard to understand by looking at Fig. 3 and 4: Significantly enriched ... are shown on the right of each profile. By looking at this pictures I see the are some dotted lines for flower and inflorescences, is that what they mean? now when I read the text it says that identification of clusters was in leaf tissue. I can't make sense of this labeling still.
I will say that the only statistical analysis that can be done in this dataset will be the enrichment text and not the differential expression given the variance and the lack of replicates. Additionally if they were to do an enrichment analysis they need to have into account this is by not means of any significant biological value, so they should also be really careful with interpretations they are making here, and should have not used Arabidopsis thaliana functional annotations but the general GO analysis.

---

## Round 0.2 · Minor Revisions

We have received two additional assessments of your work from the same previous reviewers. They both found your manuscript to be greatly improved overall. They suggested only a few additional minor revisions that will further improve the manuscript and that should be easy to integrate in the next version.

In summary they ask you to:

-Add another line to Table 1 that indicates the total number of unigenes that were assembled.

-Add one sentence in the Methods section that explains why both leaf and bud tissues were collected and what value both of these tissues added to the research question.

-try and summarize each section (except for the introduction) and to move some of the tables and the last figure (fig 5) to supplementary material.

Reviewer 1 ·

Basic reporting

Thank you for clarifying where the raw data has been uploaded and for providing the code for the analyses. This greatly improves the transparency and reproducibility of this work.

The authors have done a better job in the Discussion of only drawing conclusions based on the data available in this study. They have now revised their text in order to not overstate the results.

I would add another line to Table 1 that indicates the total number of unigenes that were assembled.

Experimental design

The authors have now provided a helpful graphic in the Supplemental materials that show what time point comparisons were made during the gene expression analyses.

I think the authors should add one sentence in the Methods section that explains why both leaf and bud tissues were collected and what value both of these tissues added to the research question.

Validity of the findings

The authors have now added (in multiple places) important caveats to their study about the lack of sufficient biological replicates. While only two replicates are available, the authors have clearly explained why only two were available and how readers should interpret the findings.

Additional comments

Thank you to the authors for making the requested changes to the manuscript. I only had a few more suggestions, but I think that this new manuscript is very much improved. The text reads much better and the Results/Discussion have been softened enough in order to not overstate the results. This will be an important preliminary work that provides new avenues for future research in the molecular pathways involved in the General Flowering phenomena in dipterocarps.

·

Basic reporting

This manuscript has address many issues I have point out in my first review comments. I understand some of the comments I made cannot be change due to the nature of the study, the limitations on sampling and the way results have been analyzed in the past. The article now reads well and has a much better understandable structure. I will encourage authors to try and summarize each section (except for the introduction) and to move some of the tables and the last figure to supplementary material, since they dont add much information to the reader (e.i. table 1)

Experimental design

I have addressed my concerns in my first review. But I recognize at this point is little what authors can do to address my comments.

Validity of the findings

I have addressed my concerns in my first review. But I recognize at this point is little what authors can do to address my comments.

---

## Round 0.3 · accepted · Accept

The final revision has addressed the additional reviewers' comments.